# Biofilm Production by Enterotoxigenic Strains of *Bacillus cereus* in Different Materials and under Different Environmental Conditions

**DOI:** 10.3390/microorganisms8071071

**Published:** 2020-07-17

**Authors:** Adame-Gómez Roberto, Cruz-Facundo Itzel-Maralhi, García-Díaz Lilia-Lizette, Ramírez-Sandoval Yesenia, Pérez-Valdespino Abigail, Ortuño-Pineda Carlos, Santiago-Dionisio Maria-Cristina, Ramírez-Peralta Arturo

**Affiliations:** 1Laboratorio de Investigación en Patometabolismo Microbiano, Universidad Autónoma de Guerrero, Guerrero 39074, Mexico; robert94a25@gmail.com (A.-G.R.); maralhi.09@gmail.com (C.-F.I.-M.); liliadiaz642@gmail.com (G.-D.L.-L.); yeezii.rs.13@gmail.com (R.-S.Y.); 2Laboratorio de Ingeniería Genética, Departamento de Bioquímica, Escuela Nacional de Ciencias Biológicas, Instituto Politécnico Nacional, Ciudad de México 11350, Mexico; valdespino_abigail@hotmail.com; 3Laboratorio de Ácidos Nucleicos y Proteínas, Universidad Autónoma de Guerrero, Guerrero 39074, Mexico; ortunoc@outlook.com; 4Laboratorio de Investigación en Análisis Microbiológicos, Universidad Autónoma de Guerrero, Guerrero 39074, Mexico; cristinasantiago81@hotmail.com

**Keywords:** *Bacillus cereus*, biofilm, enterotoxin

## Abstract

Foodborne illnesses, such as infections or food poisoning, can be caused by bacterial biofilms present in food matrices or machinery. The production of biofilms by several strains of *Bacillus cereus* on different materials under different culture conditions was determined, as well as the relationship of biofilms with motility, in addition to the enterotoxigenic profile and candidate genes that participate in the production of biofilms. Biofilm production of *B. cereus* strains was determined on five materials: glass, polystyrene, polyethylene, polyvinylchloride (PVC), PVC/glass; in three culture media: Phenol red broth, tryptic soy broth, and brain heart infusion broth; in two different temperatures (37 °C and 25 °C), and in two different oxygen conditions (oxygen and CO_2_ tension). Furthermore, the strains were molecularly characterized by end-point polymerase chain reaction. Motility was determined on semi-solid agar. The *B. cereus* strains in this study were mainly characterized as enterotoxigenic strains; statistically significant differences were found in the PVC material and biofilm production. Motility was positively associated with the production of biofilm in glass/PVC. The *sipW* and *tasA* genes were found in two strains. The results of this study are important in the food industry because the strains carry at least one enterotoxin gene and produce biofilms on different materials

## 1. Introduction

Biofilms are complex microbial ecosystems formed by one or more species of microorganisms immersed in an extracellular matrix of varied composition. Examples of microorganisms that can be found in these biofilms include bacteria and fungi [1]. Foodborne illnesses, such as infections or food poisoning, can be caused by bacterial biofilms present in food matrices or machinery [2].

*Bacillus cereus* is an aerobic or facultative anaerobic Gram-positive microorganism, endospore-forming, which can grow under different environmental conditions and in a wide range of temperatures, being resistant to heat, chemical agents and radiation [3]. The persistence of vegetative cells of *B. cereus* on food production surfaces is important in the health area and is associated with the production of biofilms. Moreover, the bacteria can survive industrial pasteurization processes due to the production of endospores. This combination complicates the removal of the microorganism by cleaning procedures [4]. This bacterial species are commonly found in industries of production of food [5,6]. In dairy products industries, *B. cereus* biofilms can be found mainly in an air-liquid interface with a typical ring attached to the walls of the tanks, from which the matrix of bacterial biofilms protrudes towards the liquid surface [7]. However, some bacterial strains are also capable of developing biofilms on submerged surfaces, for example, in stainless steel tanks or pipes [8,9].

Strains belonging to the *B. cereus* group can produce food poisoning due to the production of different toxins, among them; cereulide, a thermostable toxin, resistant to acidic pH and proteases [10]; diarrheagenic toxins, which include hemolysin BL (HBL), non-hemolytic enterotoxin (NHE), and cytotoxin K (cytK). The hbl and nhe enterotoxins are made up of three subunits: L2, L1, and B, as well as nheA, nheB, and nheC, respectively. Whereas, cytK is made up of a single protein from the member of the β-barrel family [3]. Currently, it has been described that two orthologous genes of *Bacillus subtilis* in *B. cereus* are important for the production of biofilms, *tasA*, and *sipW*; the first of them codes for a protein that forms amyloid-like fibers and the second for a peptidase that is part of the maturation of the *tasA* peptide [11].

In this study, the production of biofilms by several strains of *B. cereus* on different materials under culture conditions was determined, in addition to the enterotoxigenic profile and candidate genes that participate in the production of biofilms, as well as the relationship of biofilms with motility.

## 2. Materials and Methods

### 2.1. Study Group

The study included 11 strains biochemically identified as group *B. cereus* according to ISO 7932:2004, and the strains have which had the characteristic of producing biofilm in different proportions on the glass. For biofilm assays, only the eight strains that were molecularly confirmed were included. The strains were isolated in previous studies from the powered infant formula (B013, B030, B092, B093, B171, B186), infant food (B362), or eggshell (B326). All the strains were isolated in southern Mexico, in food distributed throughout the country. The *B. cereus* ATCC 14579 strain was used as a positive control strain for molecular identification *of B. cereus* [12], toxin gene profiling (*hbl*+) [13], motility and genes related to biofilm production. As a control of amplification of *nhe* and *cytK* genes, a strain of *B. cereus* previously characterized in the laboratory was used [14].

### 2.2. Molecular Identification and Toxin Gene Profiling of B. cereus Isolates

The chromosomal DNA was obtained by heat shock from bacterial cultures. Briefly, cells from one colony were suspended in sterile water, heated at 95 °C for 3 min, and then placed on ice. After centrifugation, the supernatants were used as a template for both the molecular identification of the *B. cereus* group, enterotoxin profile and genes that possibly participate in biofilm production.

The differentiation of *B. cereus* group was carried out by the amplification by polymerase chain reaction (PCR) to the *gyrB gene* and the toxin gene profiles from *nheABC*, *hblABD, ces*, and *cytK* gene. The final mix for each PCR reaction contained 0.2 mM of each dNTP, 3 mM MgCl_2_, 0.2 µM of the primers, 1 U of Taq DNA polymerase (Ampliqon, Odense, Denmark), 5 µL of 10X Buffer and 1 µL of DNA as a template. The PCR cycling conditions and primers are shown in Table 1. A *B. subtilis* strain was used as a negative control. In the laboratory, this strain was isolated from vegetables, biochemically identified and characterized as *B. subtilis* by amplification of the cellulase gene.

Electrophoresis was performed on 2% agarose gels at 80 V for 120 min. The gels were stained with Midori Green (Nippon Genetics, Düren, Germany) and visualized with UV light.

A 100 bp molecular weight marker was used in all electrophoresis (CSL-MDNA-100BP, Cleaver Scientific Ltd., Warwickshire, England, UK).

### 2.3. Biofilm Production

Previous to the determination of static biofilms on polyvinylchloride (PVC), coupons of PVC were placed inside glass tubes. The coupons were pretreated to remove dirt and other organic components, according to Reference [8]. The biofilms were generated in brain heart infusion broth (BHI), soybean trypticase broth (TSB), and phenol red broth (PRB). Each glass tube containing the PVC coupon was filled with 3 mL of the corresponding broth. The broths were inoculated with 1% volume of a 24-h culture (3 mL). The tubes with the PVC coupons were incubated at 37 °C and 25 °C, with and without carbon dioxide tension, from a Gaspak system for 48 h under static conditions. For the determination in glass and polyethylene; the procedure was similar, but without the PVC coupons and using tubes of each respective material. In the case of polystyrene, 96 well microplates were used, which were filled with 200 µL of the respective broths and 1% of a 24-h culture. Incubation times were the same.

Next, the biofilm formation was measured by performing crystal violet assay, as described elsewhere [16]. Briefly; after incubation, the growth was read by removing 200 µL of culture and reading it at an absorbance of 600 nm. Then, in the case of PVC, the coupons were carefully washed three times by dipping them in saline phosphate buffer (PBS) (Life Technologies, Carlsbad, CA, USA) using sterile tweezers. In the case of the glass and polyethylene tubes, as well as the polystyrene plates, the medium was removed, and they were washed three times with PBS. The adhered biofilm was stained with 0.1% crystal violet solution (BD Difco, Franklin Lakes, NJ, USA) for 30 min. The coupons, as well as the glass and polyethylene tubes and the polystyrene plates, were washed with PBS another three times, and incubated with 70% ethanol for 30 min to release the biofilm-bound violet crystal. The solubilized violet crystal was quantified by measuring the absorbance at a wavelength of 595 nm. The crystal violet tests were repeated in three independent experiments. The culture medium without inoculum was used as a negative control. To determine the specific biofilm formation (SBF), the formula proposed by Niu and Gilbert was used [17].

### 2.4. Motility Determination

Motility was determined; for this, 3.5 g/L of bacteriological agar (0.35%) was added to the respective [18] broths used for biofilm production in to use them as a semi-solid medium. One colony was stabbed and incubated under the same temperatures and oxygen conditions as those used in biofilm formation. After the incubation times had elapsed, the size of the colony was measured. The tests were repeated in three independent experiments. A *Klebsiella pneumoniae* strain was used as a negative control.

### 2.5. Determination of Genes Related to Biofilm Production in Bacillus cereus

Genes related to biofilm formation were determined by PCR amplification of the regions coding for *tasA* and *sipW* [11]. The final mix for each PCR reaction contained 0.2 mM of each dNTP, 3 mM MgCl_2_, 0.2 µM of the primers; 1 U of Taq DNA polymerase (Ampliqon, Odense, Denmark), 5 µL of 10X Buffer and 1 µL of DNA as a template. The PCR cycling conditions and primers are shown in Table 1. Electrophoresis was performed on 2% agarose gels at 80V for 45 min. The gels were stained with Midori Green (Nippon Genetics, Düren, Germany) and visualized with LED light. We performed an in silico analysis with the genome of strain *B. cereus* ATCC14579 and the primers described by Caro Astorga [11], observing the amplification of both genes of interest of the desired size, and we used strain *B. cereus* ATCC14579 as a positive control.

### 2.6. Statistical Analysis

The results represent the average of at least three independent experiments. The effects of media, material and temperature on biofilm formation by *B. cereus* strains were compared using one- way analysis of variance (ANOVA) with Bonferroni (media and temperature) and Tukey’s post hoc (material). To relate biofilm production to motility, a linear regression was performed. Statistical significance was considered when the *p* value was less than 0.05.

## 3. Results

### 3.1. Molecular Characterization of Group B. cereus Strains

To analyze the biofilm formation, we included eleven bacterial isolates identified biochemically according to ISO 7932: 2004, with the production of lecithinase, beta hemolysis in sheep blood agar and negative mannitol. For further characterization we performed molecular methods and identified eight strains belonging to the *B. cereus* group by PCR amplification of a region of the topoisomerase *gyrB* (Appendix A). This group of bacteria was also characterized by its ability to grow in different foods and surfaces, and they were positives to plcR-cytK or *nhe*, but not to *ces* genes when amplifying by PCR (Appendix A) (Table 2).

### 3.2. Biofilm Production

Next, we determined the production of biofilms in three different culture media (BHI, TSB, and PRB) and five different materials (glass, polystyrene, polyethylene, polyvinyl chloride and glass/polyvinyl chloride) at 25 °C. Significant differences were found between the three different media. For B093 strain between BHI and TSB media (*p* ≤ 0.05) on glass; BHI and TSB media (*p* < 0.0001), TSB and PRB media (*p* < 0.0001) on polystyrene. For B013 strain between BHI and TSB media (*p* = 0.002), TSB and PRB (*p* = 0.01) on glass. For B326 strain between BHI and PRB media (*p* < 0.0001), TSB and PRB (*p* ≤ 0.001) on polyethylene; BHI and PRB media (*p* ≤ 0.05) on PVC and glass/PVC (*p* ≤0.05). For B362 strain between BHI and TSB media (*p* ≤ 0.0001), BHI and PRB media (*p* ≤ 0.01) on PVC; BHI and TSB media (*p* ≤ 0.0001), BHI and PRB media (*p* ≤ 0.01) on glass/PVC.

Biofilm production was determined at a temperature of 25 °C with oxygen and compared at a temperature of 37 °C with CO_2_ using the three different culture media and the five materials. Significant differences were found for B013 on glass, and higher production was observed in BHI at 25 °C with O_2_ to 37 °C with CO_2_ (*p* ≤ 0.01). For B362, differences were found on polyethylene, PVC, glass/PVC, in the first material, a higher production was found in BHI at 37 °C with CO_2_ in relation to 25 °C with O_2_ (*p* ≤ 0.001), as well as in TSB (*p* ≤ 0.0001); in the case of PVC, higher production in TSB was found at 25 °C with O_2_ to 37 °C with CO_2_ (*p* ≤ 0.05), as well as in PRB (*p* ≤ 0.05); on glass/PVC, a higher production was observed in TSB at 25 °C with O_2_ to 37 °C with CO_2_ (*p* ≤ 0.0001), as well as in PRB (*p* ≤ 0.0001). In the case of B326, polyethylene, PVC and PVC/glass differences were found, with an increase in BHI and TSB production at 25 °C with O_2_ to 37 °C with CO_2_ in the first material (*p* ≤ 0.0001) (*p* ≤ 0.0001) and with the same effect, but with PRB for the second material (*p* ≤ 0.05) a third material (*p* ≤ 0.005). For B186, polystyrene and glass/PVC differences were found, both in TSB (*p* ≤ 0.01) (*p* ≤ 0.0001) (Figure 1).

However, when grouping the strains and comparing the production of biofilms for each strain in different materials by culture medium, we find the following: In BHI, a greater production on PVC was observed in comparison to glass (*p* ≤ 0.0001), polyethylene (*p* ≤ 0.0001) or polystyrene (*p* ≤ 0.0001). Whereas, it was observed a greater production on PVC with respect to glass/PVC (*p* ≤ 0.0001)). In PRB, a higher production was determined on glass/PVC with respect to glass (*p* ≤ 0.05), polyethylene (*p* ≤ 0.05) and polystyrene (*p* ≤ 0.05) (Figure 2).

### 3.3. Biofilm Production and Motility

It was also determined the relationship between motility and biofilm production in the same culture broths, but with bacteriological agar (semi-solid medium) at 25 °C with O_2_. We observed an increase in the production of biofilms as the size of the swarming halo increases (motility) on PVC (R = 0.5197, *p* = 0.0436) and glass/PVC (R = 0.5041, *p* = 0.0485) in TSB (Figure 3).

### 3.4. Determination of Genes Related to Biofilm Production in Bacillus cereus

Finally, Table 1 shows a summary of the genetic and biofilm formation profiles for the strains studied in this work. We observed the ability of the strains to produce biofilms in at least three materials (polystyrene, PVC, PVC/glass), which all produce PVC in all three different culture media. Moreover, all the strains have at least one enterotoxin gene, and two strains have the two genes associated with biofilm production (Figure 4), and one produces biofilm in all the materials used (B362).

## 4. Discussion

*Bacillus* strains, including strains from the *B. cereus* group can produce biofilms in different environments in the food and the beverage industry [19]. The presence of biofilms containing *B. cereus* is of great concern to the food industry (particularly in the production of fresh food, red meat, and dairy products) and as a potential source of recurrent and post-process cross-contamination in finished products, sometimes results in the decomposition of the product or food poisoning [20]. As a result, articles published about the production of biofilms by these microorganisms have exponentially increased since 2000 [19]. Although the crystal violet methodology is the most convenient method to evaluate the production of biofilms in a large group of microorganisms, including *B. cereus* [8,16,21,22,23], many experimental variables must be evaluated to evidence the capacity of strains to produce biofilms. These variables include the culture medium, temperature, incubation time, materials in which the production is induced, and the addition of nutrients or trace elements [8,21,23]. For this reason, the objective of this work was to determine the production of biofilms of *B. cereus* strains isolated from different foods under different environmental and culture conditions.

Initially, our study included eleven strains, which were biochemically identified (according to ISO 7932: 2004) with the production of lecithinase, beta hemolysis in sheep blood agar, and negative mannitol. However, only eight strains positive for *B. cereus* were confirmed by PCR amplification of the *gyrB* gene. Unlike other methodologies, such as the 16s rRNA, the *gyr*B gene amplification is advantageous because it distinguishes the strains of *B. cereus* from the strains from other genera of *Bacillus* isolated from foods, thus offering the advantage of not restricting or sequencing the gene of interest [12].

Earlier evidence by Hayrapetyan and co-workers showed that the best culture medium for the determination of biofilms in *B. cereus* was the brain heart infusion broth (BHI); however, Kwon’s group described a greater production in soybean broth trypticase [8,23]. Nevertheless, we tried phenol red broth particularly because this medium allows adding any carbohydrate, to monitor its fermentation, or even add other components to the medium. In addition, an elemental characteristic is that this broth does not disrupt the microbial density or restrict the amount of nutrients, which is important considering the production of biofilms consumes a significant number of nutrients [24]. Moreover, the advantage of phenol red as pH monitor offers an important tool, since it has been reported that the decrease in pH contributes to the production of biofilms and is related to the use of carbohydrates [23,25]. Our results showed no differences in terms of growth in the presence or absence of carbohydrates, and the growth was optimum even when the protein source was available. In the same work, Kwon et al. found differences in the production of biofilms in some strains of *B. cereus* grown at 24 or 48 h in soybean broth trypticase, suggesting that 48 h is optimum for developing biofilms. Nevertheless, we found no significant differences between distinct culture media and *B. cereus* strains, except for some cases, indicating that any medium could be used for this determination.

It has been described previously that the production of biofilms is lower on polystyrene or glass microplates compared to stainless steel (8). In the case of glass, one explanation is related to a possible low amount of extracellular DNA, which has been described as an adhesin that favors the interaction with this material, and therefore, the production of biofilms [26]. Although a high production was found in polyethylene tubes, PVC was the material where a high production of biofilms was observed. We noticed that in the literature, only one report works with *B. cereus* and PVC microplates; however, they did not compare with other materials and only related the adhesion and the presence of the S layer with this material [4]. Although in this study the methodology proposed by Castelijn et al. for PVC was followed [16], using PVC and immersing them in broth contained in glass tubes, we observed that many of these strains managed to produce biofilm in a glass tube containing PVC, but not in the glass tube alone. For this reason, we decided to include PVC as a material, observing significant differences and greater production of biofilms. This phenomenon may be related to the fact that biofilms once mature at 24 h could release spores or vegetative cells into the medium, which could spread, and in this case, adhere to the glass tube, which has been described [9]. The other possible hypothesis is related to the fact that during the production of biofilm on PVC, DNA is released into the extracellular medium, and this favors the adhesion of the cells to the glass (since this is considered as an adhesion requirement) [26]. It is important to denote the use of this material (PVC) in the study, due to the few data that exist for *B. cereus* and its usefulness in the food industry. This is because polyvinyl chloride (PVC), low-density polyethylene (LDPE), and rubber are FDA approved food contact substances that are widely used as important components of conveyor belts [27]. In the case of temperature, Kwon’s group works with 25 °C and 30 °C, showing that there is a greater production of biofilms at 30 °C, which is also consistent with those reported by Wijman and co-workers [9,23]. In our study, we decided to work with 25 and 37 °C, considering the first one as optimal temperature for the development of *B. cereus* and the second one as body temperature in which it causes disease. We observed statistically significant differences in various materials, particularly for four strains. The differences between strains have been evidenced in multiple studies, all-coinciding in the metabolic fitness and particular characteristics of each strain [8,9,23].

It has been shown that the flagellum does not participate in the adhesion of bacteria to glass [28]. However, there are four possible mechanisms that could mediate the motility of this microorganism by the peritrichous flagella participates during the production of the biofilm. First, motility is a key element for the formation of biofilms when the bacterium has to search suitable places for its formation, so that inhibition of motility results in the formation of submerged biofilms. Second, motile bacteria create channels in the matrix, which increase the entry of nutrients. Third, motile cells can enter the biofilm and increase biomass. Fourth, motile bacteria located on the edge of the growing biofilm can migrate and favor the spread of the colony or biofilm [19]. Therefore, in this study, we considered important to analyze the participation of the flagellum indirectly, using the motility technique on semi-solid agar (not shown). Among the most important observations was the relationship between the increase in the swarming halo and the formation of biofilms on PVC and glass/PVC (Figure 3). It could in part, help us to understand the phenomenon described above, in which, in addition to producing biofilms in PVC, they produced in the glass that contained PVC; confirming the hypothesis of several studies, where the motile bacteria helps colonization or search for new materials for the production of biofilms from a preformed biofilm [19,24].

It should be mentioned that the importance of these *B. cereus* strains identified in this study is that all of them are capable of producing at least one enterotoxin, but not cereulide. In this sense, two events related to the enterotoxigenic capacity of the strains have been described; the first one is when they are in the biofilm and become a focus of contamination or defined as a toxic patch, which would be releasing enterotoxigenic strains that would contaminate the food [7]. On the other hand, Caro-Astorga’s group describes two possible subpopulations of cells, those that are capable of producing biofilms and the enterotoxigenic ones, which are released during erosion or dispersion [24].

Moreover, Caro-Astorga et al. have described the participation of different genes in the production of biofilm, such as the *eps* operon [24], and the *sipw* and *tasA* genes [11]. In this sense, another important aspect in the study is the presence of genes recently related to biofilm production; *tasA* production has been associated with the formation of amyloid-like fibers that are responsible for the floating biofilm in *Bacillus* species. In the case of *sipW*, it encodes for a protease that participates in *tasA* [11] processing, hence the importance of finding them. Even though the frequency is low, it is the first study that does it descriptively and relates it to different aspects of biofilm production, finding that a strain with both two genes is capable of producing biofilms in all the materials used.

## Figures and Tables

**Figure 1 microorganisms-08-01071-f001:**
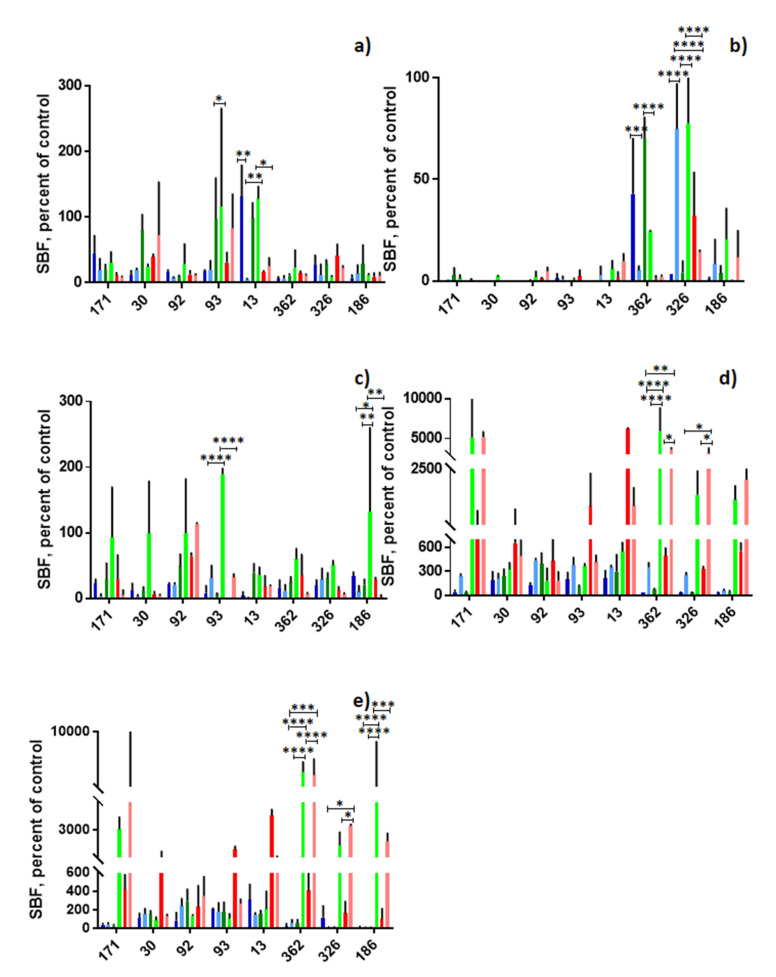
Biofilm production in different culture media, materials, oxygen conditions and temperature. (**a**) Glass; (**b**) polyethylene; (**c**) polystyrene; (**d**) PVC; (**e**) PVC/glass. Blue bars: BHI at 25 °C with O_2_, light blue bars: BHI at 37 °C with CO_2_, green bars: TSB at 25 °C with O_2_, light green bars, TSB at 37 °C with CO_2_, red bars: PRB at 25 °C with O_2_, pink bars: PRB at 37 °C with CO_2_. * *p* ≤ 0.05, ** *p* ≤ 0.01, *** *p* ≤ 0.001, **** *p* = 0.0001. The dotted line denotes the absorbance of the negative control. The standard error is graphed in black. In the X axis are the strains and in Y the absorbance at which the biofilms were read.

**Figure 2 microorganisms-08-01071-f002:**
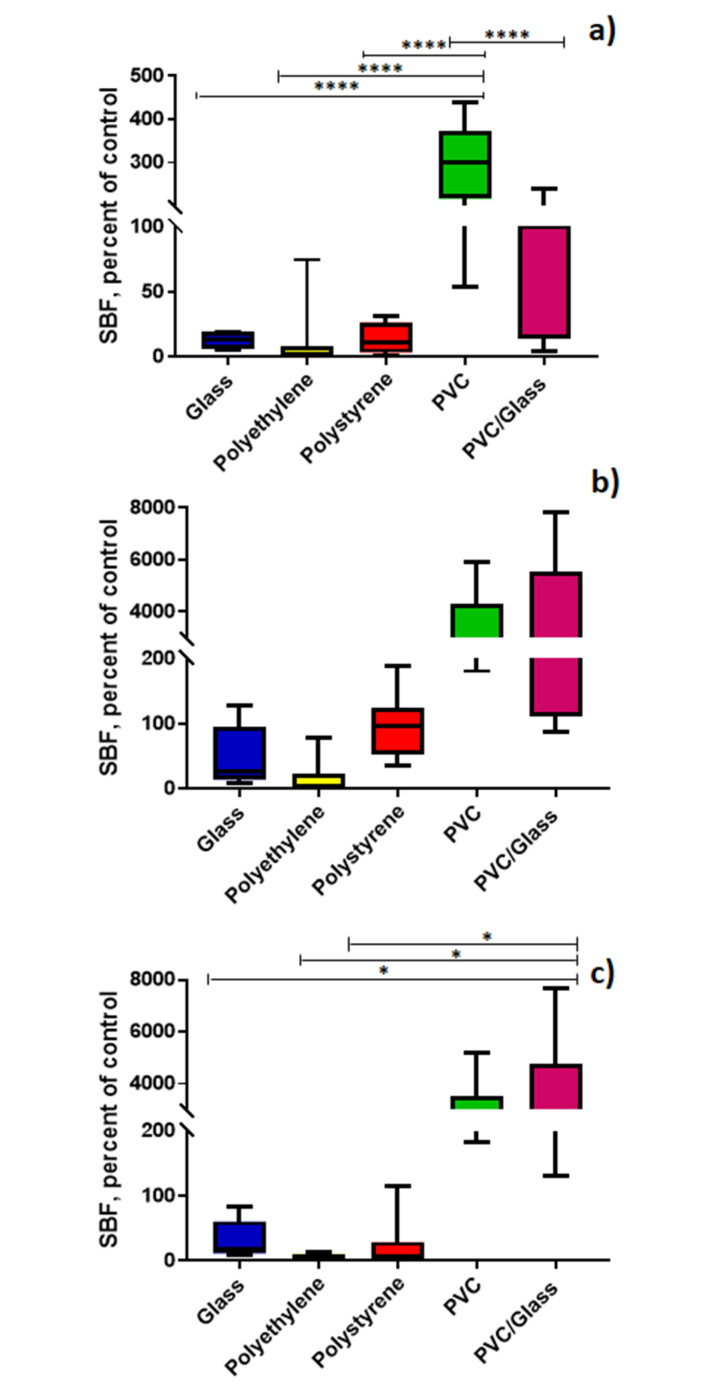
Biofilm production in different culture media and materials at 25 °C with O_2_. (**a**) BHI; (**b**) TSB; (**c**) PRB. * *p* ≤ 0.05, **** *p* ≤ 0.0001. In the X axis are the materials and in Y the specific biofilm formation (SBF).

**Figure 3 microorganisms-08-01071-f003:**
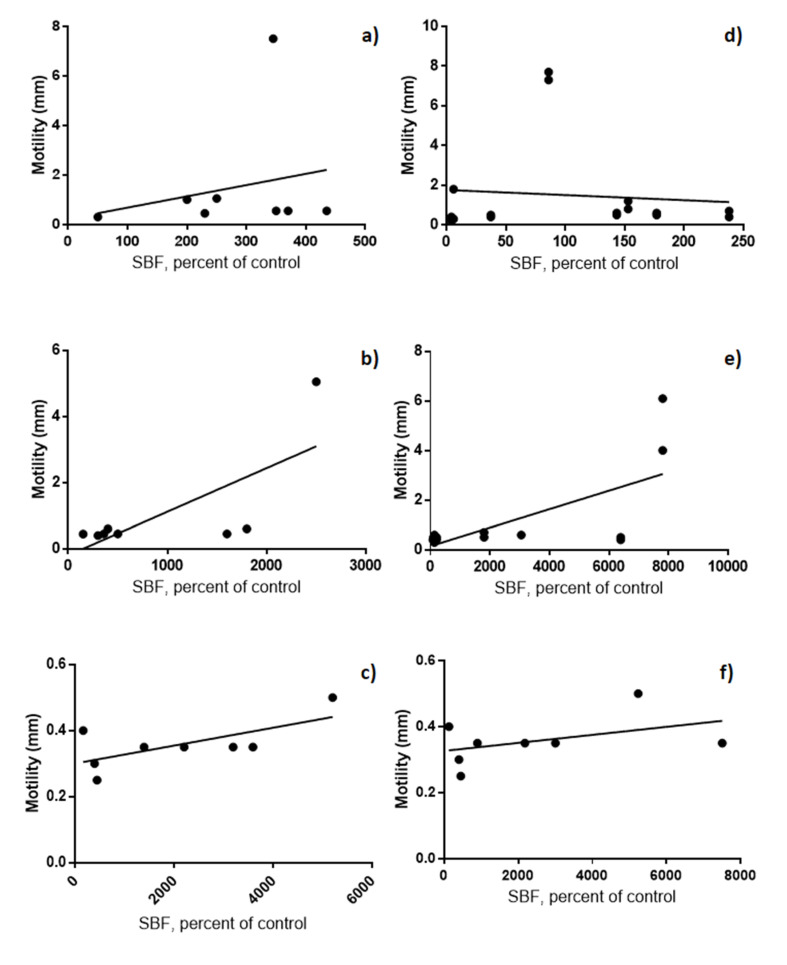
Biofilm production and motility at 25 °C with O_2_. (**a**–**c**) PVC; (**d**–**f**) PVC/glass; (**a**,**d**) PRB; (**b**,**e**) TSB; (**c**,**f**) BHI. The X axes show the absorbance detected using the crystal violet biofilm assay and the Y axes show the swarming halo diameter.

**Figure 4 microorganisms-08-01071-f004:**
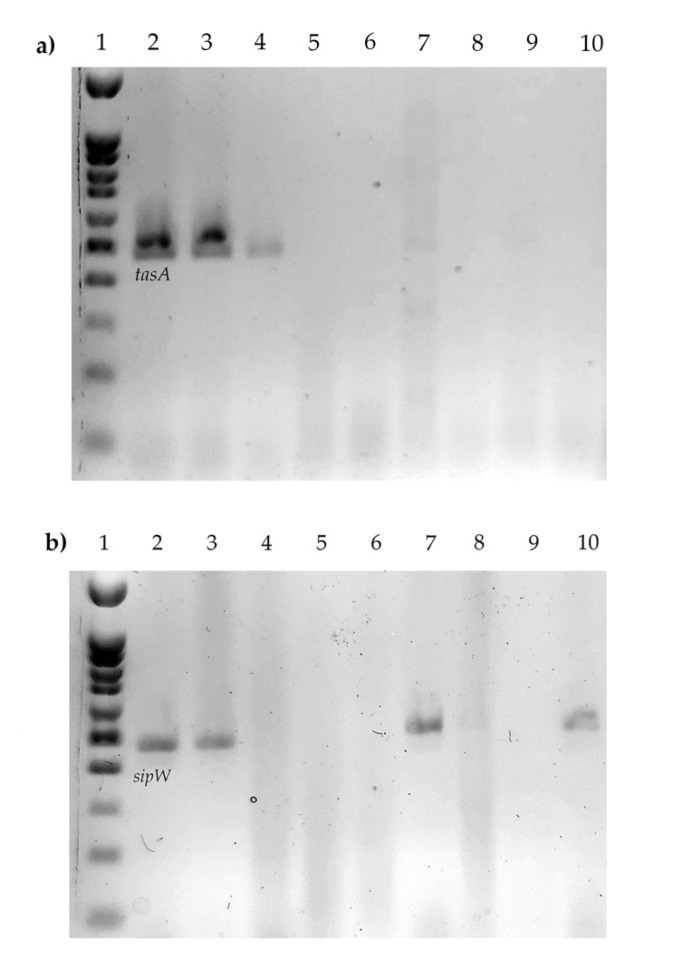
*tasA* and *sipW* genes. (**a**) *tasA*, 1. Molecular weight marker, 2. B133, 3. B030, 4. B326, 5. B013, 6. B092, 7. B093, 8. B171, 9. B186, 10. B362. (**b**) *sipW*, 1. Molecular weight marker, 2. B133, 3. B030, 4. B092, 5. B013, 6. B093, 7. B326, 8. B186, 9. B362, 10. B171.

**Table 1 microorganisms-08-01071-t001:** Polymerase chain reaction cycling condition and primer sequences.

Gene	Primer Sequences	PCR Cycling Conditions	Reference
*gyrB*	F- GCC CTG GTA TGT ATA TTG GAT CTA C R- GGT CAT AAT AAC TTC TAC AGC AGG A	Initial denaturation of 2 min at 94 °C, followed by 30 cycles at 94 °C for 30 s at, 52 °C for 1 min and 72 °C for 30 s, and final elongation at 72 °C for 10 min.	[12]
*nheABC*	F- AAG CIG CTC TTC GIA TTC R- ITI GTT GAA ATA AGC TGT GG	Initial denaturation of 5 min at 94 °C, followed by 30 cycles at 94 °C for 30 s, 49 °C for 1 min and at 72 °C for 1 min, and final elongation at 72 °C for 5 min	[13]
*hblABD*	F-GTA AAT TAI GAT GAI CAA TTT C R- AGA ATA GGC ATT CAT AGA TT		
*ces*	F- TTG TTG GAA TTG TCG CAG AG R-GTA AGC GGA CCT GTC TGT AAC AAC	Initial denaturation of 2 min at 94 °C, followed by 30 cycles at 94 °C for 30 s at, 52 °C for 1 min and 72 °C for 30 s and a final elongation at 72 °C for 10 min	[13]
* cytK-plcR *	P1- CAA AAC TCT ATG CAA TTA TGC AT P3- ACC AGT TGT ATT AAT AAC GGC AAT C		[15]
* tasA *	F- AGC AGC TTT AGT TGG TGG AG R-GTA ACT TAT CGC CTT GGA ATTG	Initial denaturation of 5 min at 94 °C, followed by 40 cycles at 94 °C for 30 s, 59 °C for 45 s and 72 °C for 45 s, and final elongation at 72 °C for 5 min	[11]
* sipW *	F- AGA TAA TTA GCA ACG CGA TCTC R- AGA AAT AGC GGA ATA ACC AAGC	Initial denaturation of 5 min at 94 °C, followed by 40 cycles at 94 °C for 30 s, 54 °C for 45 s, and 72 °C for 45 s, and a final elongation at 72 °C for 5 min	

**Table 2 microorganisms-08-01071-t002:** Biofilm production, enterotoxigenic profile and biofilm-related genes in *Bacillus cereus* strains.

Strain	Source	Biofilm					Enterotoxin Profile *	*sipW **	*tasA **
		Glass	PS	PE	PVC	PVC/Glass			
**13**	**PIF**	**b, c**	**b, c**		a, b, c	a, b, c	*nhe, cytk*	-	-
30	PIF	a, b, c	b		a, b, c	a, b, c	*hbl, nhe, cytk*	+	+
92	PIF	b	a, b, c		a, b, c	a, b, c	*nhe*	-	-
93	PIF	a, b, c	a, b, c		a, b, c	a, b, c	*nhe*	-	-
171	PIF	b	b		a, b, c	a, b, c	*nhe, cytk*	+	-
186	PIF		b		a, b, c	b, c	*nhe, cytk*	-	-
326	eggshell		a, b	a, b	a, b, c	b, c	*hbl, nhe, cytk*	+	+
362	Infant food	b	b	a, b	a, b, c	a, b, c	*nhe*	-	-

(a) BHI, (b) TSB, (c) PRB. * determinate by endpoint PCR. PIF, powered infant formula; PVC, polyvinylchloride.

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
