# Peer review of "Biofilm Production by Enterotoxigenic Strains of Bacillus cereus in Different Materials and under Different Environmental Conditions"

_microorganisms, 2020, doi:10.3390/microorganisms8071071_

Round 1

Reviewer 1 Report

This paper by Roberto et al describes characterization of a set of B. cereus strains with regard to genetic complement, biofilm formation on various materials, and motility, using multiple media and temperatures.    The data appear solid, though they do not show any clear pattern of genetics, motility, and biofilm formation.  If such a pattern could be established, then it might be useful for the development of methods to combat contamination of food processing equipment by this potential pathogen.

While the work appears adequately done, the manuscript is much in need of improvement.  Clarity in the presentation of data in figures and tables is required for several sets of data.  A great deal of editing for English language use is needed for clarity.  The discussion is quite long and is unclear throughout.

Specific comments:

  1. Line 24: This should say "on" five materials. In all cases throughout the manuscript, biofilms are formed "on" materials like glass, not "in" glass.
  2. Line 37 should say “Biofilms are complex microbial ecosystems”
  3. Line 38: “different” should be “varied”
  4. Line 49: This is an odd list, as dairy and beverage would be included within "food"
  5. Line 68: Define "ISO"
  6. Lines 88-107: There is a lot of repetition in these different PCR assays. It seems like a single protocol could be stated, followed by the slight variations for each gene.
  7. Lines 108-109 are identical to 85-86. Delete one set.
  8. lines 126-127 should say, "The medium was removed"
  9. Line 134: The phrase "to relate it to the production of biofilms" is not needed here in the Methods sections.
  10. Line 136: It is not clear what "pitting" means.
  11. Lines 141-151: Again, too much repetition.
  12. Figures 1, 2, and 6 are very large and provide very little information for the reader to use. They are also summarized in Table 2/ These figures should be
  13. moved to supplementary data.
  14. Lines 170-175: Specify the temperature: 37 or 25 degrees?
  15. Does Figure 5 contain some of the exact same data as figure 3, just graphed now with the 37 degree data?
  16. Line 211: Should 30 be 25 degrees?
  17. Lines 217-223: This entire section on the relationship between motility and biofilm production is unclear, and the corresponding methods section does not make it clear. I am guessing that the values for each strain were plotted on a graph with X-axis being biofilm and Y-axis being motility and a line was drawn for these points for each medium and substrate. In this case, a graph might actually be useful, showing the relationships between motility halo and biofilm production, and the lines for which R values are being calculated.
  18. Line 231: What is the basis for saying that this strain "produces them in all of the materials used"? First: Produces what? The toxins or the Biofilm proteins? Second: How was production of these specific proteins measured? Is this based simply on the production of biofilms by this strain? As we have no evidence  that these proteins are actually present in those biofilms, this statement is not supported.
  19. The discussion is very long. The first paragraph is review of material from the introduction or of previously published material. This paragraph can be deleted or reduced to one sentence.
  20. The discussion needs a great deal of editing, by a very strong English-language editor, for clarity, with attention to being brief and to the point. There are run-on sentences, unclear references to "this material" (several times), and very unclear statements about results and conclusions.
  21. Line 274: Either there were or were not significant differences. One can't add "except in some cases." What does that mean?
  22. Line 280: It is not clear what "this material" refers to.

Reviewer 2 Report

Line 21. Please, change  “under culture” by “under different culture”

Line 26. Please, change CO2 by (CO2)

Line 28. Please, ad a point between “agar” and “The”

Lines 31-33. Please, rewrite this sentence. ¿Why the strains in this study are important in the food industry? May be, the results of this study may be important to understand…

Lines38-39. Please, rewrite. The definition of biofilm is independent of the environment where it is produced. They are not only produced depending on the type of food that a food industry produces. Please, separate the common definition of biofilm from the presence of biofilms in the food industry.

Line 48. ¿These bacterial species? Or This bacterial species. Authors are talking about only one species. B. cereus. Please, rewrite.

Line 54. Please, change “Some strains of group B. cereus can produce” by “Strains belonging to the B. cereus group can produce”.

Line 63-65. There is a correlation between the presence of some genes and biofilm formation, but correlation does not imply causation. To demonstrate that a gene causes a phenotype, this gene must be mutated and the phenotype must be lost. And this is not the case.

Line 71-72. ¿what is the clonal relationship between the strains? Phylogenetic data are missing to understand the link between these strains.

Lines 80, 113, 123 and so on. Please, please avoid using “as described previously or previously described”. This is only used when the reference is from the same group that writes the paper. The formula should be used is: “as described elsewhere”.

Line 115. Please, indicate the initial volume of the O/N culture.  1% of how much volume? 10 ml, 100ml, 10.000L ??

Line 133. This reviewer misses a negative motility control and a positive motility control.

Line 134. please indicate the% agar in the text (0.75%). Is this type of motility in B. cereus, swarming?

151-152. This reviewer misses a statistics section, to avoid putting the comments on statistics in the figure captions.

Line 162. This reviewer misses a positive and a negative control for PCR.

Line 162. Figure 1. Please, use the same marker as in figure 6. Figure 1 is of poor quality.

Line 162. Figure 1. Bands seem too bad for a 2% gel.

Please, indicate what are the markers (name, molecular weight and brand).

Line 165. Figure 2. Poor quality.

Line 217. Please, provide images of the motility plates.

Line 235. Figure 6. Poor quality.

Comment: Please use the same marker on all gels, the bands are less than 1000 bp in all cases, ¿why change?. This is good for homogenizing images.

Lines 256-258. Please, rewrite.

Lines 261, 262 and so on. ¿who is (8)? ¿who is (20)? are they persons ? Are they research groups? Are they the work of other researchers? Please quote appropriately, for example: Johnson and coworkers… In the work of… A recently published work… The results obtained by…

Specific Comments to authors.

Biofilm formation in vitro. Authors should consider normalizing the biofilm development with the growth, which is common practice, to account for growth rate differences. This is mandatory, because, B. cereus can form biofilm under different growth conditions but for most strains, these environmental conditions also heavily influence the growth rate, and this paper shows very different environments (temperature, CO2…).

Authors need to use a criterium to classify biofilm former strains. i.e.: Christensen et al. If OD exceeded 0.24, classified the strain as strongly adherent. If the OD is less than or equal to 0.12, classified the organism as nonadherent.  Also, you can introduce other criteria such as: non-biofilm-former, weak biofilm-former, moderate biofilm-former and strong biofilm-former.

For instance, the Authors used a PCR approach to study the presence of some genes. These genes are not the only genes associated with biofilm formation. In some studies other genes have been demonstrated to play a major role in biofilm variation.

At least one control (positive/negative) to run experiments must be included (PCR, biofilm formation, motility...).

The choice of 30ºC or 37ºC as an incubation temperature doesn't seem very obvious in the discussion section, and this may in fact be one of the more important aspects of the study related to the food industry.

Round 2

Reviewer 1 Report

The paper still requires some moderate editing for English grammar in addition to some specific changes I recommend below.

Line numbers refer to the pdf version of the paper that showed all tracked changes.

Line 21 should say “on” different materials.

Line 39 should say “ecosystems”

Line 65 should say “on” different materials.

Line 187: “isolations” should be “isolates”

The sentence on lines 284-286 is very unclear.  It needs to be shortened and simplified, and possibly split into 2 sentences.

Line 286: “increasing” should be “increase”.  Actually, this sentence makes no sense. Increase in biofilm production relative to what?  More motility correlates with more biofilm?

New figure 3:  The legend is unclear. I am guessing the X-axes represent the relative amount of biofilm measured for each strain. The axes should be labeled “Biofilm (A at 590nm)” and the legend should say “The X axes show the absorbance detected using the crystal violet biofilm assay and the Y axes show the swarming halo diameter (or radius?)”

Line 319 should say “crystal violet”

Line 350 should say “on” polystyrene.

Line 364 should say “on” PVC.

Line 379: I do not know what “peritric” means.

Lines 386-387 should say, “Among the most important observations was the relationship between the increase in the swarming halo and the formation of biofilms on PVC and glass/PVC”

Reviewer 2 Report

Please, read the attached document. 
